# A Study on the Ultrasonic Regulation of the Welding Performance and Residual Stress of 316L Stainless Steel Pipes

**DOI:** 10.3390/ma15186255

**Published:** 2022-09-08

**Authors:** Xiaowei Jiang, Chunguang Xu, Jingdong Li, Jiangquan Lu, Lin Wang

**Affiliations:** 1Jiangsu Institute of Automation, Lianyungang 222061, China; 2Key Laboratory of Fundamental Science for Advanced Machining, Beijing Institute of Technology, Beijing 100081, China; 3College of Mechanical Engineering, Inner Mongolia University of Technology, Hohhot 010051, China

**Keywords:** ultrasonic wave, 316L stainless steel, residual stress, temperature field

## Abstract

Due to its extreme service conditions, low-temperature pressure piping often needs post-welding stress measurement and control. Aiming at the phenomenon of local stress concentration in welded 316L pipes, this study used ultrasound to regulate the stress in the welded area at different times during and after the multi-layer welding of the pipeline butt joint for different time lengths. Mechanical properties such as tensile strength and hardness were tested for each comparison group, and the microcrystalline phases of the weld and its surrounding microstructure were analyzed. The transverse and longitudinal surface residual stresses of each comparison group were measured. The influence of high-energy ultrasound on the surface temperature field during and after welding was analyzed. The experimental results show that ultrasonic wave regulation can speed up heat exchange and radiation in the weld zone (WZ), refine the grains in the WZ, heat-affected zone (HAZ) and fusion zone (FZ) to some extent and reduce and homogenize residual stress to a certain degree. In the 120 mm area of the weld center, the residual stress measured after the mid-welding regulation was smaller than that of any other comparison group. This regulation result was the best, followed by that of hot regulation and finally that of offline regulation. The tensile strengths obtained by the mid-welding regulation and post-welding hot regulation of this group were the best, increasing by 17.2% and 24.3%, respectively, compared with the untreated groups.

## 1. Introduction

In industrial fields such as marine, shipping and nuclear power, 316L low-temperature pressure piping is often faced with fatigue failure, stress corrosion, fractures and other problems [1,2,3,4,5,6,7,8,9,10]. To obtain good molding performance and high weld microstructure properties and to reduce the stress concentration caused by welding deformation, international scholars have put forward many effective methods [11,12] for mid-welding and post-welding treatments. Mechanical stretching, vibration aging, stress-relieving annealing, post-welding surface impacts, rolling, welding with trailing ultrasonic impacts, high-energy ultrasonic regulation and other methods have been used to reduce stress concentration [13,14,15,16,17,18,19,20,21,22,23,24,25,26,27]. After the studies that used above methods, it was known that all the processes of eliminating residual stress were processes of transforming elastic strain into plastic strain, either by reducing the yield strength of the material to turn elastic strain into plastic strain, or by using plastic strain to offset elastic strain [19,28]. In the 1960s, French scientist Langenecker [29] was the first to discover the softening effect of ultrasound on solid metal materials, which could reduce the yield strength of the materials to a certain extent, transform part of the elastic strain into plastic strain and finally relieve residual stress. On this basis, scholars later developed ultrasonic vibratory stress relief (UVSR), ultrasonic impact treatment (UIT), the ultrasonic cavitation process (UCP) and other methods, and they studied the regulation effect of those methods on the microstructures and stress in different materials to further verify their effectiveness.

For 316 stainless steel, Shalvandi et al. [30] compared the results of residual stress elimination by thermal stress relief and by UVSR. The results showed that the stress reduction by UVSR was 36%, close to that by thermal stress relief (40%). Zhang et al. [23] used UVSR to reduce the stress in 6082 aluminum alloy plates and found that the average stress relief rate was up to 57% and that local yield stress was reduced by at least 27% during UVSR. Through simulation analysis, they found that the rate of residual stress relief was directly proportional to the ultrasonic frequency response intensity introduced to the area. D.A. Lesyk et al. [31] studied the surface roughness, residual stress and microstructure of 304 stainless steel after multi-pin UIT. Compared with the initial condition, the surface roughness (Ra) was reduced by about 91.1%. Mechanical surface treatment caused a significant reduction in the grain size of sample austenite (15–20 nm) and martensite (20–37 nm), which further brought about nature hardening, grain boundary hardening and dislocation hardening. The effects of ultrasonic frequency and ultrasonic application time on the material‘s surface, microstructure and residual stress were studied in the above studies by different ultrasonic application methods compared with conventional methods. However, the stress or microstructure regulation effect and the action mechanism of ultrasonic waves in dynamic processes such as the welding process, the post-weld thermal state and the post-weld cooling state of the samples still have not been studied.

Some scholars also studied the effect of ultrasound on melted or cooled solids to improve the microstructure properties according to the effect of ultrasound on the physical state of welding. Dehnavi et al. [32] found through studies that ultrasonic sound pressure also had a tensile and compression effect on molten pools, and that the ultrasonic action on melting could produce some nonlinear acoustic phenomena such as acoustic cavitation, which could significantly increase the thermal conductivity of the liquid weld pool and improve grain size, tensile strength, microhardness and wear resistance. Cui et al. [33] found that the ultrasonic waves introduced into the molten pool through solids caused ultrasonic cavitation and fluidization in the mixture zone, which promoted the complete mixing of fused filler metal and base metal at the freezing front and improved the utilization efficiency of ultrasonic energy. Qiang et al. [34] studied the influence of arc ultrasound on the post-welding properties of the MGH956 diffusion-strengthened alloy and found that the frequency and amplitude of the excitation current were closely related to the porosity and tensile strength of the post-welding structure. The above studies have further broadened the timing and scope of ultrasonic applications, whose coverage, however, is still insufficient. In addition, the differences in physical states between weld beads and that of ultrasonic application time were not considered [20].

Although many studies have been carried out in recent years on the application of ultrasound in solid and molten metal materials, most of them are still limited to the single properties of specific samples, such as yield stress, residual stress and material microstructure, without comprehensive research on the mechanisms of specific stress regulation, stress distribution and multi-physical-field coupling, and mechanical properties and microstructure properties for the multi-layer bead of a tubular structure. Therefore, the foundation for further large-scale applications of this technology in general manufacturing fields such as welding and machining cannot be laid.

In order to deeply study the influence of ultrasound on the multi-layer welding process of a 316L pipe and on its post-welding microstructure properties and residual stress, ring-shaped array ultrasonic waves were applied to base metal in this study by changing the regulation duration in the offline mode and the regulation timing in the online mode. The mid-welding and post-welding regulation of welds were carried out. The microstructure properties, mechanical properties and stress relief results that were obtained with different treatment methods were studied and compared. The law of temperature field distribution in different treatment methods was analyzed to further reveal their residual-stress relief mechanism. This study provides a technical basis for realizing the low stress manufacturing of stainless steel butt joints through gas tungsten arc welding (GTAW) with high efficiency and low cost.

## 2. Experimental Section

### 2.1. Material

The base metal (BM) used in the test was a 316L butt-welded seamless stainless steel pipe with a 168 mm diameter, an 8 mm wall thickness and a 600 mm length(Figure 1). Double V grooves (35°) with root faces were opened on the butt ends. A 1.2 mm GMS-316L solid wire containing Mo was used to improve the creep performance at high temperatures and to improve resistance to pitting corrosion in a halide atmosphere. The main chemical composition of BM and welding wire is shown in Table 1. Argon of 99.99% purity was used as the protection gas. During welding, the interlayer cooling time was 5 min, and the interlayer temperature was strictly controlled below 150 °C.

### 2.2. Test System and Test Method

The test system in this paper consisted of an automatic robot welding system, an ultrasonic wave system and a flexible clamping system. The robot welding system used a KUKA KR16 six-axis industrial robot (KUKA, Augsburg, Germany) and a Fronius Trans Tig 5000 Job G/F TIG power(Fronius, Welles, Austria) supply. The ultrasonic wave system used a NYK5887-L6 power(Mingke Electromechanical Co., Ltd., Ningbo, China) amplifier with continuous output power up to 800W and an output frequency range of 100 Hz–100K Hz. In the system, a RIGOL DG1022 dual-channel function/arbitrary waveform generator(Puyuan Jingdian Technology Co., Ltd., Beijing, China) was used, with the highest sampling frequency of 100MSa/s. Six groups of circumferential array high-energy ultrasonic exciters were designed. The ultrasonic transducer waws a PZT4 piezoelectric ceramic transducer(Dawei ultrasonic equipment Co., Ltd., Shaoxing, Zhejiang) with a diameter of 70 mm and a resonant frequency of 14.64 kHz. The distance between the transducer and the weld center could be adjusted. To facilitate the comparison of regulation effects, all the transducers in this study were arranged on side A, 130 mm away from the weld center, as shown in Figure 2. The flexible clamping system included a U-shaped caliper positioner for self-centering clamping and high-precision rotational positioning, an external track for the robot and a supporting roller. The general diagram of the welding test system is shown in Figure 3.

The system was designed with two high-energy ultrasonic excitation schemes, namely a mid-welding online scheme and a post-welding offline scheme, as shown in Figure 3, to facilitate the analysis of ultrasonic impacts under mid-welding and post-welding excitation. The mid-welding online ultrasonic excitation was a high-energy ultrasonic excitation mode during welding or under the post-welding hot condition, whereas the post-welding offline ultrasonic excitation was a stress regulation mode for the offline welded pipe that had been cooled down.

### 2.3. Test Method

As the GTAW used in this study had a small filling capacity, the four-layer bead technique was selected to complete the backing welding, filler welding and cosmetic welding of the bead. The relevant process parameters are shown in Table 2. The method of mechanical high-precision fixed-length rotary cutting was used to ensure that the dimensional and geometric tolerances of the pipe met the butting requirements. The butt gap of the welds was 1 mm, and the single-side groove was 35°. All the welds in the four-layer bead started at the same point on the circumference of the pipe joint, as shown in Figure 4. Before welding, the oxidation film and oil stains on the base metal surface were removed by mechanical grinding and acetone wiping. A high-temperature couplant was applied between the transducer tail and the wedge and between the wedge and the pipe to be welded to ensure effective coupling of the structures and to realize the effective output and introduction of ultrasonic waves during and after welding.

To study the regulation effect of ultrasonic waves on the residual stress, mechanical properties and microstructure of 316L stainless steel, five groups of welding samples, as shown in Figure 5, were designed for a comparative test, and the ultrasonic parameters were set, as shown in Table 3. The first group, defined as U_0_, was conventional gas metal arc welding(GMAW) without the application of ultrasound. The second group, defined as U_1_, was GMAW followed by 1 h of cooling and then 1 h of the application of high-energy ultrasound (U-GTAW-C1). The third group, defined as U_2_, was GMAW followed by 1 h of cooling and then 2 h of the application of high-energy ultrasound. The fourth group, defined as U_3_, was the 10 min regulation of each of the 2–4 bead layers during welding, with 0.5 h of the application of ultrasonic excitation. The fifth group, defined as U_4_, was post-welding hot regulation, i.e., the 10 min regulation of each of the 2–4 hot layers for a total of 0.5 h. Among them, the group U_0_ was without ultrasonic application, the groups U_1_ and U_2_ were with offline ultrasonic application and groups U_3_ and U_4_ were with online ultrasonic application, as shown in Figure 6.

To ensure the validity of the comparison sample data, the 0° starting points and circumferential 330° points of the above five groups of welding samples were selected as the sampling center points of tensile specimens and metallographic specimens. The shape and specifications of the specimens are shown in Figure 7. X-rays were used to detect the internal defects of the samples, which were then cut by WEDM to obtain the specimens after no defects were detected. The surfaces of the metallographic specimens were polished and corroded, and then they were microstructurally observed by OLYMPUS GX51 microscopy(OLYMPUS, Tokyo, Japan). The tensile test was carried out at the speed of 3 mm/min based on ISO4136:2022 by using a SANS-XYB305C tensile testing machine(SANS testing machine co.,LTD, Shenzhen, China). The hardness test was carried out by a Q750 Qness hardness tester(Qness, Salzburg, Austria) according to ISO 6507-1-2018 and was based on Qpix T12 software(Qness, Salzburg, Austria) for automatic image analysis and hardness measurements. For the temperature field test, a Fluke TiX580 thermal infrared imager(Fluke, Washington, USA) was used to check the distribution of the temperature field on the surface of the pipe during welding or in a hot state, and it was used to draw the temperature gradient curve.

## 3. Results and Analysis

### 3.1. Microstructure Analysis

The grain morphology of the five groups of weld samples is shown in Figure 8. The group U_0_ showed the microstructure of the weld zone without ultrasonic treatment, which was basically composed of long and coarse columnar crystals growing from the weld fusion line to the center. The groups U_1_ and U_2_ show the microstructures obtained after 1 h and 2 h for post-welding ultrasonic regulation, which revealed no obvious influence of ultrasonic regulation on the weld grains. The group U_3_ and U_4_ showed the microstructural morphology under the mid-welding regulation and post-welding hot regulation. It can be seen that, under the action of ultrasound, the size of columnar crystal zones was significantly reduced, the columnar crystal roots along the radial and axial directions were clearly refined and equiaxed dendritic crystals existed between the columnar crystal zones.

By comparing U_3_ and U_4_, it can be seen that, under the mid-welding regulation, the columnar crystal zone of the weld was the least obvious, and the proportion of the equiaxed dendrite crystals was the largest. This is because the application of high-energy ultrasound in the welding process brought the acoustic flow effect [35] and ultrasonic cavitation effect [36,37], which accelerated the molten pool flowing and thermal convective circulation; therefore, the columnar crystals were broken and taken away from the fusion line zone by the melting. The above phenomena provide the possibility for the nucleation of equiaxed fine crystals [38].

The microstructures of PMZs (partially molten zones) of different comparison groups are shown in Figure 9. In the PMZ zone of group U_0_, the matrix was composed of austenites, and many large chain-shaped δ ferrites were found at the grain boundary. In groups U_1_ and U_2_, under the offline regulation, the large chain-shaped δ ferrites were reduced due to the introduction of high-energy ultrasound. In groups U_3_ and U_4_, under the online regulation, the large ferrites were uniformly refined and dispersed to the austenite grain boundary for precipitation, and the grains were clearly refined.

Through comparative experiments, it was found that the online regulation of high-energy ultrasound can play a role in refining the crystals in the melting zone. Due to the effect of high-energy ultrasound, the initial ferrite dendrites increase; therefore, more dendrites spread to the crystal boundary to refine the microstructure. However, under offline regulation or cold regulation, the internal stress brought by the application of high-energy ultrasound stimulates the dislocation source action in the grains, resulting in slip and then elastic deformation release and macroscopic plastic deformation. As a result, the grains are not substantially refined.

High-energy ultrasound also has an impact on the morphology of columnar crystals near the weld fusion zone. As shown in Figure 10, the columnar grains of groups U_0_, U_1_ and U_2_ were coarse with wide boundaries. However, the grains of groups U_3_ and U_4_ were smaller with narrower boundaries and had radially decomposed equiaxed dendrites between the columnar crystals. The equiaxed dendrites in U_3_ had the largest number and the smallest size. This is because the mid-welding ultrasonic regulation of U_3_ and the post-welding hot ultrasonic regulation of U_4_ could accelerate the heat exchange inside and outside the microstructure and decrease the internal temperature quickly to a more balanced point; therefore, the grain refinement degree was better than that without ultrasonic treatment.

### 3.2. Tensility

The tensile test was carried out on the five groups of experimental samples. As the tensile strength of the BM and HAZ was much higher than that of the WZ zone, all tensile specimens were fractured at WZ. The experimental curves are shown in Figure 11. The yield inflection points occurred at the strains of 0.010–0.015. The yield limits of elastic zones corresponding to the inflection points of the five groups of curves were different; the yield limit of U_4_ was the highest, the yield limits of U_2_ and U_3_ were close to each other and the yield limits of U_0_ and U_1_ were basically the same. The slopes before the inflection points were basically the same. The above results show that online ultrasonic regulation can obtain better linear elasticity by extending the elastic zone.

It can be seen from the tensile strengths and elongations in Table 4 that the suboptimal tensile strength of 606.2 MPa and the optimal elongation of 35.64% were obtained by the mid-welding ultrasonic regulation of U_4_, increasing by 17.2% and 92%, respectively, over the untreated comparison group U_0_. However, the optimal tensile strength of 643.2 MPa and the suboptimal elongation of 35.03% were obtained by post-welding hot ultrasonic regulation, increasing by 24.3% and 88.7%, respectively, over U_0_. This further confirms that the strength of polycrystals increase with grain refinement degree. Based on the Hall–Petch formula [39], it can be understood that, as the grain size becomes smaller, the yield strength becomes greater.

### 3.3. Hardness

A total of 15 points were sampled from the sample groups U_0_–U_5_ along their BM, HAZ and WZ zones, and then their hardness was measured by a Vickers hardness (HV) tester. The comparison results are shown in Figure 12. The hardness distribution of the samples in each group almost followed a similar law, i.e., the hardness of BM was the highest, followed by WZ and HAZ. High energy ultrasound had a small impact on BM but a great impact on WZ and HAZ.

The 0.5 h mid-welding regulation of U_3_ had the greatest impact on the heat-affected zone (HAZ), greatly increasing the HAZ hardness by 9.6% on average from the pre-treatment 151 HV to post-treatment 163–168 HV. The HAZ hardness of U_4_ (0.5 h post-welding hot regulation) increased by 4.6% on average from the pre-treatment 151 HV to post-treatment 157–159 HV. The HAZ hardness of the other two comparison groups U_1_ and U_2_ was somewhat improved, but this impact was small. The coarse columnar crystals in the HAZ were refined due to the influence of ultrasound, thus causing changes in hardness [31].

Groups U_1_ and U_2_ were subjected to post-welding cold regulation, which had little effect on their WZ hardness. Their WZ hardness was generally similar to that of the control group. It can be seen from the above that the offline regulation had no obvious effect on either grain size or hardness.

The effect of the two methods of online regulation (U_3_ and U_4_) on HAZ was higher than that of the two methods of offline regulation, which indicates that the online regulation can substantially improve the mechanical properties of the material.

### 3.4. Temperature Field

The distribution of the surface temperature fields of U_0_ and U_4_ at 5 min after welding and that of U_3_ during online welding are shown in Figure 13. As can be seen from the surface temperature field of U_3_, the mid-welding temperature of the ultrasonic introduction side was significantly higher than that of the other side at the far end, and the temperature of the area near the transducer was higher. This was caused by the accumulation of heat without timely exchange due to the continuous heat input during welding. The energy of the high-energy ultrasonic field and the heat field was superposed in that area, causing the accumulation of heat on the local surface near the transducer and the high temperature of that area. However, a normal thermal gradient took place at the far end.

Compared with the control group U_0_, group U_4_ followed a completely opposite trend of temperature field distribution without local temperature superposition. The temperature of the area near the ultrasonic excitation side was generally lower than that on the right side. This is because only the ultrasonic field, rather than double field sources (welding heat source and ultrasonic field source), existed at that time. The existence of the ultrasonic field sped up heat exchange and environmental radiation; therefore, the heat in the area near the ultrasonic field could be quickly exchanged.

The above-mentioned hot regulation and mid-welding regulation had different impacts on temperature fields and different stress regulation mechanisms. The former was to bring about local temperature rise through the coupling and superposition of heat fields and high-energy ultrasonic fields in the welding process, and the latter was to speed up heat exchange and radiation by the use of high-energy ultrasonic fields and then to improve the microstructural crystallization.

### 3.5. Residual Stress

The residual stress ultrasonic detector meeting the national standard GB/T 32073-2015 was used to measure the transverse and longitudinal residual stresses of groups U_0_–U_4_ on sides A and B 20 mm away from the weld center, as compared in Figure 14. As can be seen from the reference group U_0_, the residual stresses on sides A and B of the pipe weld circumference were basically symmetric. The amplitudes of transverse and longitudinal residual stresses changed significantly along the circumference, which was related to weld starting position, heat input, circumferential constraint and other welding factors. Under any of the four regulation schemes U_1_–U_4_, the overall stress-relieving effect of the excitation side A was better than that of side B. The online regulation schemes U_3_ and U_4_ had the best effect on circumferential residual stress relief and homogenization. The offline regulation schemes U_1_ and U_2_ also worked, but U_2_ had a much better stress-relieving effect than U_1_.

The above phenomenon appeared because the use of ultrasonic waves for online regulation improved the microstructural crystallization, refined the grains and improved the stress condition. The offline regulation caused the transformation of elastic strain in the material into plastic strain and the release of residual stress. However, since the regulatory energy attenuated with the increase in the propagation distance, there was an obvious residual stress gap between sides A and B.

The transverse and longitudinal distribution of the axial residual stress is shown in Figure 15. Similarly, any of the ultrasonic regulation methods had an influence on the transverse and longitudinal residual stresses. Among them, U_3_ (mid-welding regulation) had the most significant stress-relieving effect, followed by U_4_ and then U_1_ and U_2_. From the point of view of offline regulation time, the stress reduction by U_2_ was greater than U_1_ but was not proportional to the time. This indicates that a longer regulation time can improve the effect of offline stress regulation, which, however, stops changing significantly after a certain time. Moreover, it can also be found that the stress reduction was relatively significant in the area near the weld, indicating that the regulation effect of high-energy ultrasound in the near-field region was significantly better than that in the far-field region. This effect was basically zero in the area beyond 120 mm.

## 4. Conclusions

In this study, ultrasound was applied at different times and durations before and after the welding process of 316L stainless steel to adjust the cooling and crystallization effect of the microstructure to achieve homogenization and reductions in residual stress in the weld. Through the comparison and analysis of the grain morphology of PMZ and weld fusion lines, the influence of ultrasound on the microstructure was studied. Tensile and hardness tests were carried out on five groups of comparison samples to further analyze the influence of ultrasound on the mechanical properties of the material. This study explored the effect of ultrasound on temperature fields in the microstructural crystallization process and investigated the distribution law of the surface temperature field along the weld circumference of the pipes under online regulation (U_4_,U_5_). The research conclusions can be drawn as follows:
(1)Through the acoustic flow effect and ultrasonic cavitation effect, the use of ultrasonic waves for online regulation transformed the columnar crystals in the weld fusion zone into equiaxed fine crystals, increased the initial ferrite dendrites in the melting zone, generally refined the grains and improved the stress condition. However, in the use of ultrasonic waves for offline regulation, the internal stress brought by ultrasonic wave propagation stimulated the dislocation source action in the grains, resulting in slip and then elastic deformation release and stress relief. In general, the stress-relieving effect of online regulation was much better than that of offline regulation. In addition, the regulation effect of high-energy ultrasound in the near-field region was significantly better than that in the far-field region but became basically zero in the area beyond 120 mm.(2)Compared with the unregulated material, the introduction of ultrasonic waves into the online (mid-welding and post-welding hot) regulation obtained better tensile strength and elongation, which could increase by more than 80% and more than 17%, respectively. Moroever, the best grain refinement effect was obtained by online ultrasonic regulation, and a higher material surface hardness was also achieved by the grain boundary strengthening effect.(3)The two modes of online ultrasonic regulation (mid-welding regulation and post-welding hot regulation) could effectively improve the HAZ hardness of the welded material by 9.6% and 4.6%, respectively, compared with that of the untreated group. However, the offline regulation had no effect on hardness.(4)During online ultrasonic regulation, the “stirring” effect of the sound field could speed up heat exchange and radiation. When the mid-welding regulation was applied, the temperature gradient in the near-sound-field region was smaller than that in the far-field region due to the superposition of the welding heat field and ultrasonic field and the formation of a local high-temperature area. When the post-welding regulation was applied, the temperature gradient in the near-sound-field region was larger than that in the far-field region.


## Figures and Tables

**Figure 1 materials-15-06255-f001:**
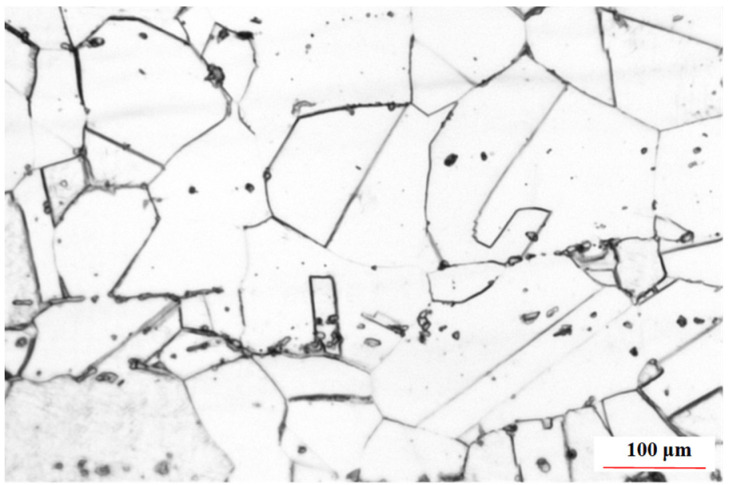
Base metal microstructure.

**Figure 2 materials-15-06255-f002:**
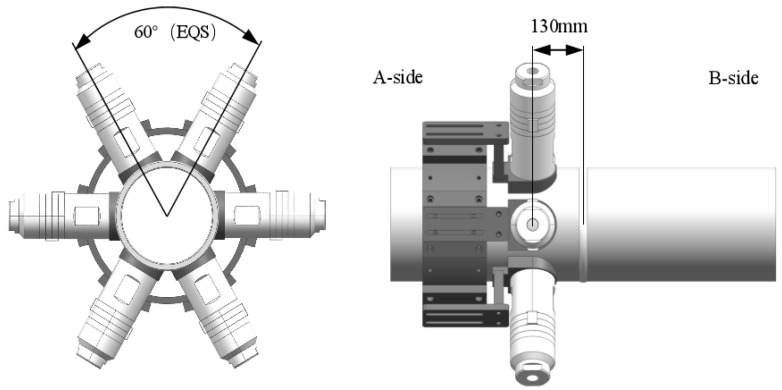
Schematic diagram of ultrasonic stress regulation scheme layout.

**Figure 3 materials-15-06255-f003:**
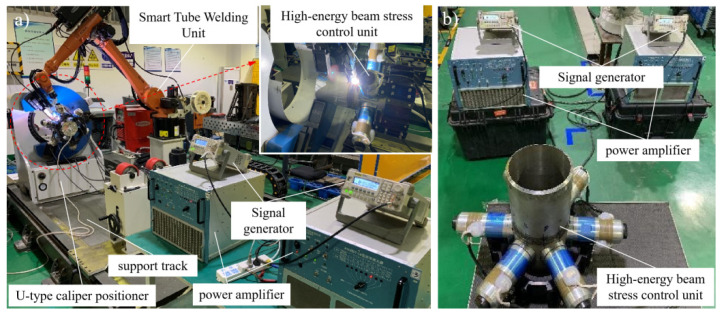
Schematic diagram of welding and stress regulation system: (**a**) Mid-welding online regulation; (**b**) Post-welding offline regulation.

**Figure 4 materials-15-06255-f004:**
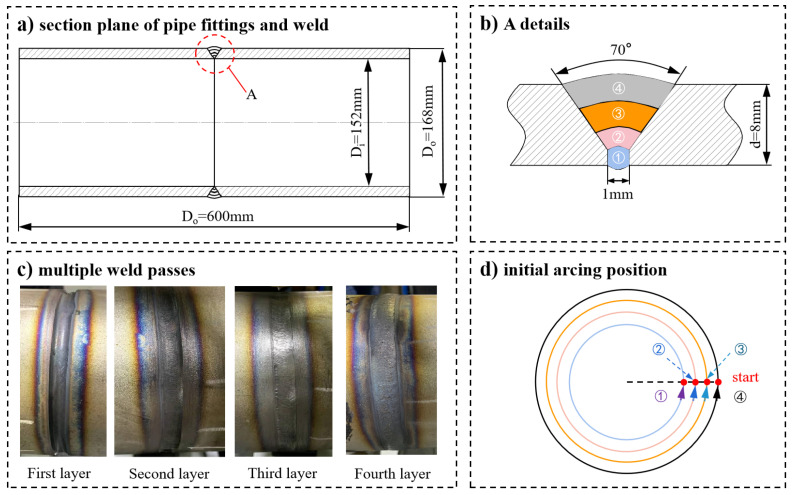
Weld bead interlayer process parameters.

**Figure 5 materials-15-06255-f005:**
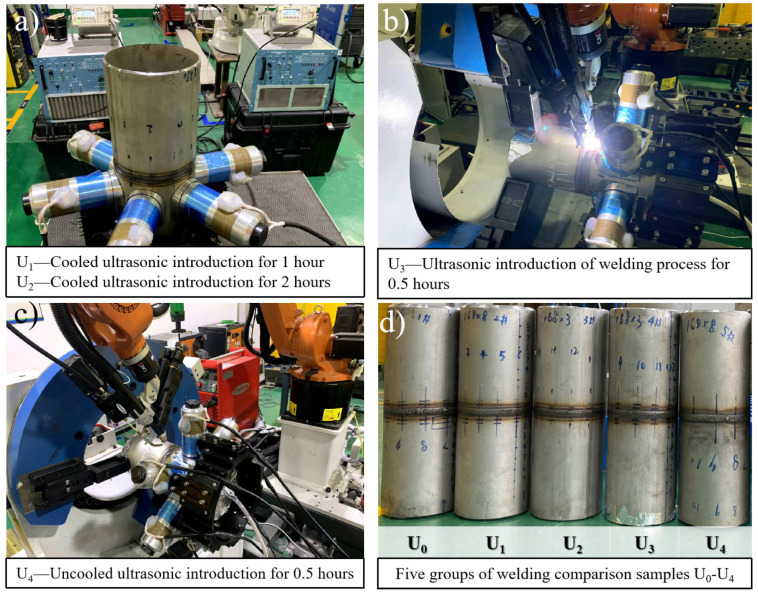
Ultrasonic welding stress regulation process and samples.

**Figure 6 materials-15-06255-f006:**
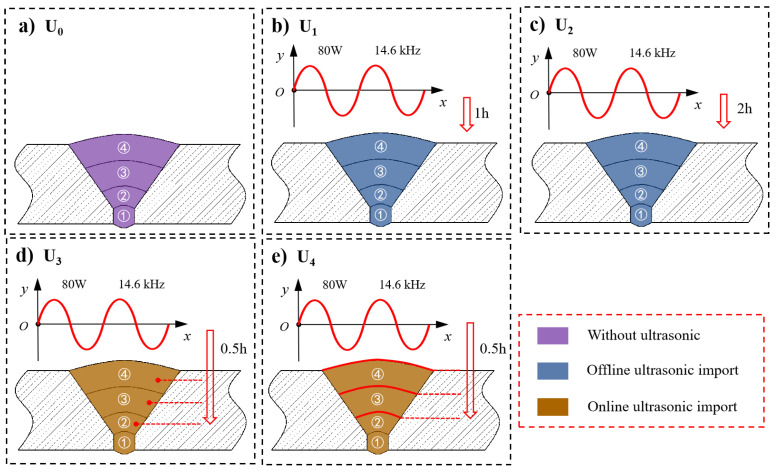
Scheme of the comparison group: (**a**–**e**) U_0_–U_4_, respectively.

**Figure 7 materials-15-06255-f007:**
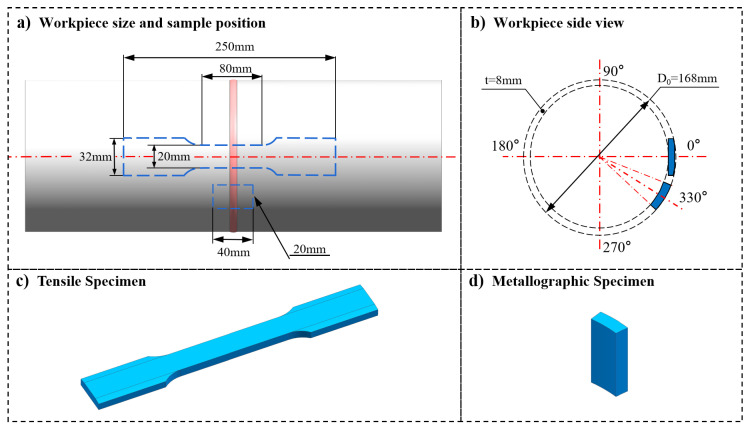
Test sampling location and size specification.

**Figure 8 materials-15-06255-f008:**
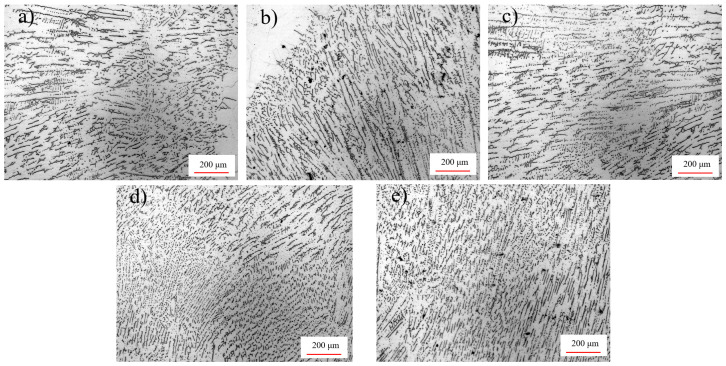
Weld grain morphology: (**a**–**e**) U_0_–U_4_, respectively.

**Figure 9 materials-15-06255-f009:**
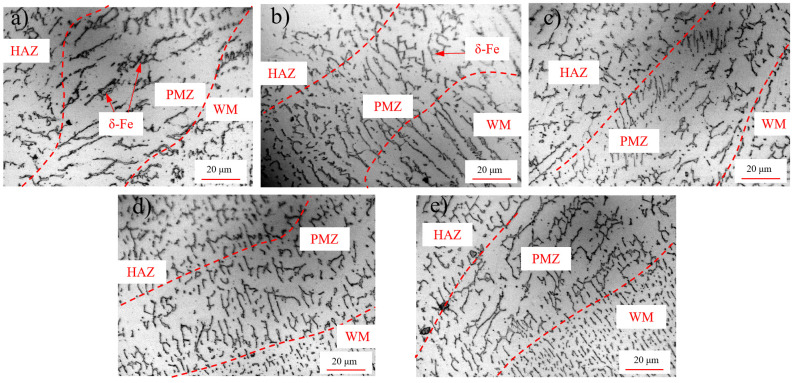
Morphology of the partially melted zone (PMZ) of the weld: (**a**–**e**) U_0_–U_4_, respectively.

**Figure 10 materials-15-06255-f010:**
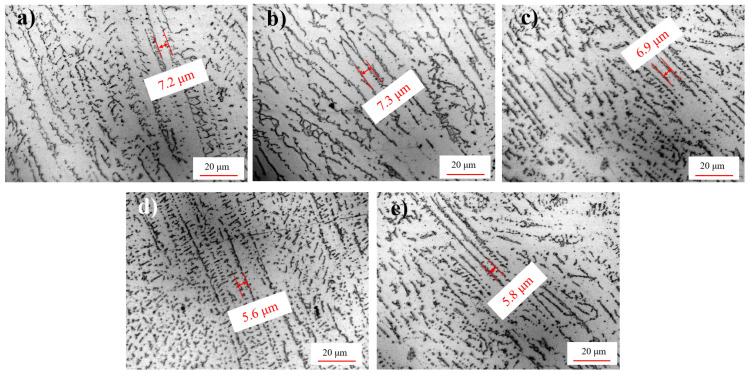
Columnar crystal morphology of weld fusion line: (**a**–**e**) U_0_–U_4_, respectively.

**Figure 11 materials-15-06255-f011:**
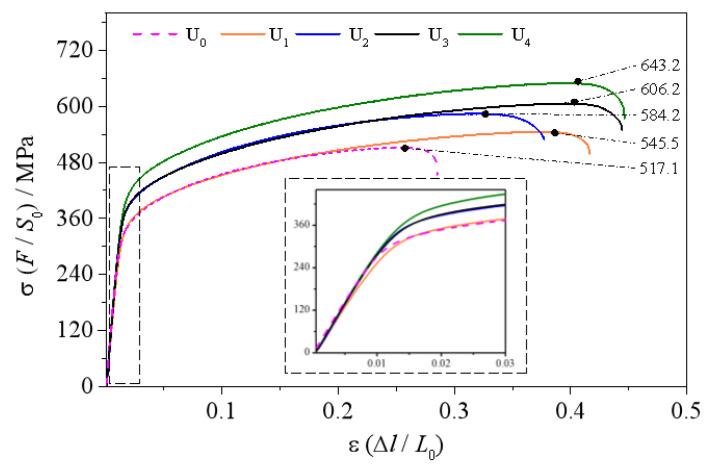
Tensile comparison curves of different samples.

**Figure 12 materials-15-06255-f012:**
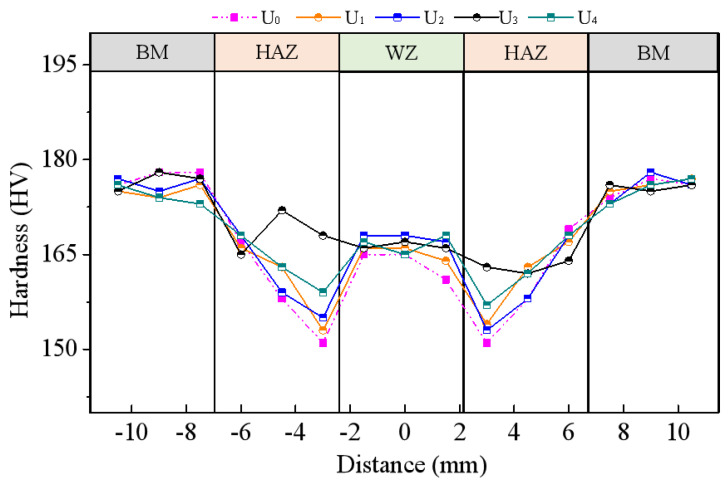
Microhardness distribution in a welded joint.

**Figure 13 materials-15-06255-f013:**
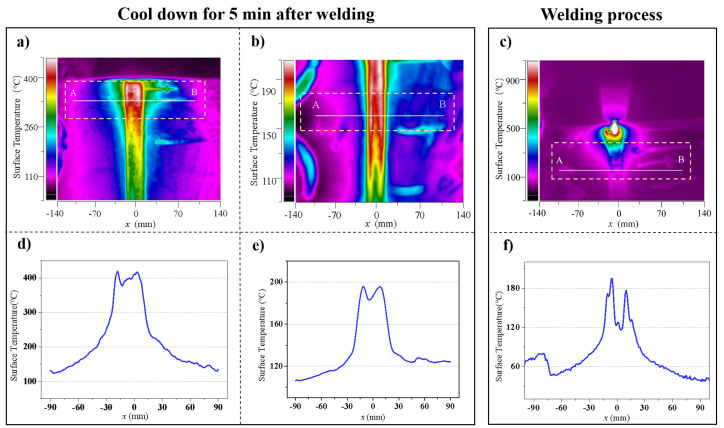
Surface temperature field distribution: (**a**–**c**) temperature field distribution of the U_0_, U_4_ and U_3_ welds, respectively; (**d**–**f**) temperature variation trend along line AB of the U_0_, U_4_ and U_3_ welds, respectively.

**Figure 14 materials-15-06255-f014:**
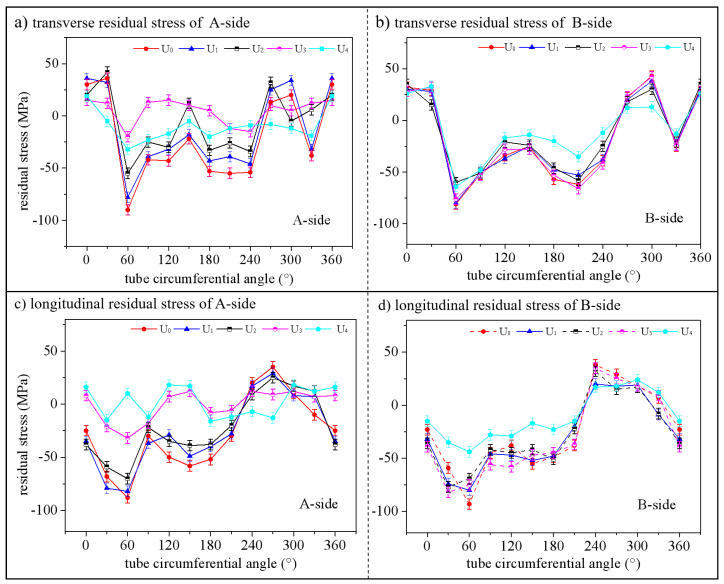
Comparison of residual stress at 20 mm from the center of the weld.

**Figure 15 materials-15-06255-f015:**
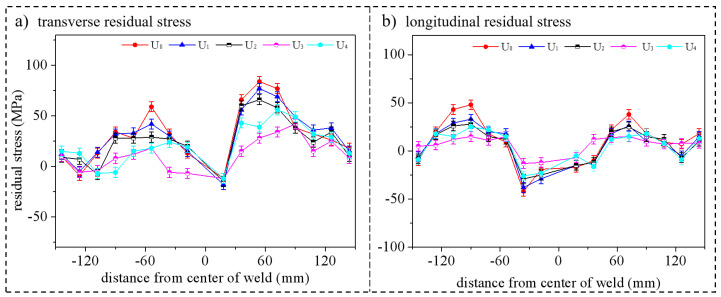
Comparison of surface residual stress along the pipe axis.

**Table 1 materials-15-06255-t001:** Chemical composition (wt %) of base material 316L and welding wire ER316L.

Material	C	Mn	Si	S	P	Cr	Ni	Mo	Fe
Base material	0.025	1.19	0.64	0.011	0.015	17.13	12.57	2.12	Bal.
Welding wire	0.025	1.91	0.42	-	-	19.10	12.58	2.57	Bal.

**Table 2 materials-15-06255-t002:** Welding process parameters.

Weld Bead	DC/Pulse	Current (A)	Voltage (V)	Welding Speed (m/min)	Swing Type	Swing Parameter
1	DC	130	40	0.06	—	—
2	Pulse	145	40	0.054	Triangular	1.2 mm
3	Pulse	145	40	0.054	Triangular	1.2 mm
4	Pulse	150	40	0.054	Triangular	1.2 mm

**Table 3 materials-15-06255-t003:** Ultrasonic parameters.

Parameters	Value
Ultrasonic frequency (kHz)	14.6
Ultrasonic power (W)	80
Ultrasonic transducer diameter (mm)	70
Number of ultrasonic transducers	6

**Table 4 materials-15-06255-t004:** Comparison of tensile parameters of samples.

Comparison Group	Max. Force (Fm/KN)	Tensile Strength (Rm/MPa)	Original Gauge Length (Lo/mm)	Final Gauge Length (Lu/mm)	Elongation (A)
U_0_	77.1	517.1	70	82.99	18.56%
U_1_	82.42	545.5	70	93.2	33.14%
U_2_	88.31	584.2	70	89.46	27.80%
U_3_	91.27	606.2	70	94.95	35.64%
U_4_	96.86	643.2	70	94.52	35.03%

## Data Availability

The data that support the findings of this study are available from the corresponding author.

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
