# Peer review of "A Study on the Ultrasonic Regulation of the Welding Performance and Residual Stress of 316L Stainless Steel Pipes"

_materials, 2022, doi:10.3390/ma15186255_

Round 1

Reviewer 1 Report

This paper deals with the ¨ Study on Ultrasonic Regulation of the Welding Performance 2 and Residual Stress of 316L Stainless Steel Pipes¨. The manuscript topic is interesting, but there are comments for improvements to the paper:

1- The abstract is too long.

2- The abstract should represents scope and results of manuscript. Please remove description of 316L steel from abstract.

3- The introduction needs to be improved. Welding of 316L Stainless Steel Pipes is almost well known topic. It is better improve the manuscript from Ultrasonic and Residual Stress topics. There are a couple of literature in this topic that need to report in Introduction.

4- The benefits and drawbacks of this research are not clear. The authors aimed for what they wanted to present, but they did not mention what kind of problem they wanted to solve.

5- Please match the citation style with Materials style.

6- The scale bars of figures are too small please revise them (Like Fig. 1).

7- Please revise numbering in Fig. 3 with alphabet (1=a and 2=b).

8- Discussion of results are not good enough. Please improve discussion section of results.

Reviewer 2 Report

This paper describes the impact of ultrasonic treatment on the microstructure and mechanical properties of 316 weldment.  Different from other studies, this paper investigates the impact of sonication on the layered (multiple welding) structure.  This paper finds the results that are consistent with what has been expected that is that 1) grain structure of weldment is refined when ultrasonic (US) treatment is applied while welding power is applied, 2) plasticity of weldment increases with US treatment (online has more impacts than the offline), 3) hardness of weldment and HAZ is generally lower than the base metal.  All results are within the expectation except for one fact, that is that hardness of weldment is lowest with U0 (no treatment) while highest with U4 (online).  This is opposite to what was expected and may be related to the grain boundary strengthening.  A bit more elaborated explanation would be great. 

Minor corrections: 1) UVSR (in introduction) is introduced without definition.

2) better define the meaning of "the dynamic welding process" in line 69-71 of introduction.

Reviewer 3 Report

The manuscript "Study on Ultrasonic Regulation of the Welding Performance and Residual Stress of 316L Stainless Steel Pipes" has been reviewed. It deals with an experimental investigation on AISI 316L steel and ultrasound utilization to regulate the stress in the welding area during and after the multi-layer 15 welding of the pipeline butt joint. The concept is interesting and new.

However in my opinion it can be reconsidered for publication after the following major revisions:

Line 109: metal, not MENTAL.

Fig. 3: a) and b), not 1) and 2).

Fig. 4: add a)... d) markers in the pictures and captions. Check "positionr" in fig. 4 d).

Fig. 6: add in captions a)... e).

Line 169; Table 2. Please specify motivation for ultrasonic parameters selection.

Line 192: standard ISO 4136:2012 is not up to date. Please check.

Line 194: standard ISO 6507-1:1982 is not up to date. Please check.

Fig. 7: add in captions a)... d).

Fig. 8: add in captions a)... e).

Fig. 9: add in captions a)... e).

Fig. 10: add in captions a)... e).

Table 4: add measurement units.

Fig. 13: add in captions a)... c).

Fig. 14: add in captions a)... d).

References (All) are not in compliance with the journal requirements.

Line 323: [179]? Please check.

Line 322: the principle of residual-stress detection should be summarized.

My final concern is about the hardness in the HAZ (usually higher than the MZ; in the manuscript is the opposite) and the tensile strength in the molten zone (usually higher then the BM). Authors are required to give an explanation for that. Also for the fracture in the welds!

Reviewer 4 Report

1.     Is each welding pass start and end points are same? What is their impact of them on the residual stress of the pipe?

2.     It is recommended to provide the heat input values to correlate/estimate the stresses in the welds.

3.     Please quantify the number of phases of the welds to withstand the stresses and strength estimations.

4.     Conclusion point 1 needs to be revised with key results instead of discussing general points.

5.     Please provide the deformation of the welds concerning the original surface.

6.     The correlation studies between ultrasonic output signals and welds physical changes and stress distribution.

Round 2

Reviewer 1 Report

The authors revised manuscript properly 

Reviewer 3 Report

The manuscript has been significantly improved and can now be accepted in the present form.

Reviewer 4 Report

Revised version is improved.